# The Impact of Mixing Techniques on PMMA Bone Cement Subjected to Two Different Cooling Techniques: A Pilot Study of Thermal Management Strategies in Orthopedic Applications

**DOI:** 10.3390/biomedicines13123071

**Published:** 2025-12-12

**Authors:** Gergo Tamas Szoradi, Andrei Marian Feier, Octav Marius Russu, Sandor Gyorgy Zuh, Tudor Sorin Pop

**Affiliations:** 1Doctoral School, George Emil Palade University of Medicine, Pharmacy, Science, and Technology of Targu Mures, 540142 Targu Mures, Romania; gergo.szoradi@umfst.ro; 2Department M4 Clinical and Surgical Sciences, Orthopedics and Traumatology I, George Emil Palade University of Medicine, Pharmacy, Science, and Technology of Targu Mures, 540142 Targu Mures, Romania; octav.russu@umfst.ro (O.M.R.); sandor.zuh@umfst.ro (S.G.Z.); tudor.pop@umfst.ro (T.S.P.); 3Department of Orthopaedics and Traumatology, Clinical County Hospital of Mureș, 540139 Targu Mures, Romania

**Keywords:** PMMA bone cement, thermal necrosis, compressive strength, hand mixing, vacuum mixing

## Abstract

**Objectives:** Polymethyl methacrylate (PMMA) bone cement is vital for prosthetic fixation in orthopedic surgery, yet its exothermic polymerization can exceed 80 °C, surpassing the 50 °C threshold for thermal osteonecrosis, risking implant failure. This pilot study assesses two cooling strategies—precooling cement components and saline irrigation on the polymerization temperature and compressive strength of antibiotic-loaded PMMA, comparing hand mixing (HM) and vacuum mixing (VM) to optimize thermal management while preserving mechanical integrity in controlled settings relevant to orthopedic applications. **Methods:** Antibiotic-loaded Simplex bone cement (Stryker, Kalamazoo, MI, USA) was prepared using HM and VM, per ISO 5833. Each batch was divided into three groups: control, precooled (components at 6 °C overnight), and saline irrigation (8 °C saline during setting). Each group included 20 cylindrical samples (1.5 cm × 3 cm), cured for 24 h. Core temperatures were monitored with embedded thermometers, and compressive strength was measured in megapascals (MPa) using a hydraulic press (C092-06, MATEST). Welch’s *t*-test was used for statistical analysis. **Results:** HM controls reached 76.2 °C, precooled 63.6 °C, and saline 66 °C; VM controls hit 71.8 °C, precooled 58.8 °C, and saline 63.6 °C. HM strengths were 16–17 MPa, with precooling reducing to 16.49 MPa (*p* = 0.051) and saline maintaining 17.07 MPa (*p* = 0.820). VM strengths were 76–80 MPa, with precooling at 78.45 MPa (*p* < 0.001) and saline at 76.77 MPa (*p* = 0.010). Failure modes varied: controls (uniform cracking), precooled (shear failure), and saline (mixed cracking/crumbling). **Conclusions:** Precooling significantly lowers temperatures but compromises strength in HM samples, limiting its use in load-bearing applications. Saline irrigation offers moderate thermal control while preserving mechanics, particularly in HM, suggesting a viable strategy for reducing thermal necrosis risk. VM ensures superior strength, supporting safe cooling application.

## 1. Introduction

Polymethyl methacrylate (PMMA) bone cement has been a cornerstone of orthopedic surgery since its introduction by Sir John Charnley in the 1970s, revolutionizing prosthetic fixation in procedures such as total hip and knee arthroplasties [1,2]. Initially developed to enhance screw fixation in bone, PMMA’s ability to provide robust mechanical interlocking with cancellous bone has significantly improved implant stability, extending the lifespan of cemented implants from approximately 10 years in early applications to over 25 years with modern techniques [3,4,5].

PMMA bone cement is a critical material in orthopedic surgery, widely used for prosthetic fixation in total hip and knee arthroplasties and for stabilizing osteoporotic vertebral compression fractures in vertebroplasty. Its exothermic polymerization, however, generates temperatures exceeding 80 °C, surpassing the 50 °C threshold for thermal osteonecrosis [6,7,8]. This heat-induced bone cell death at the bone–cement interface can create necrotic zones, trigger inflammatory responses, and promote fibrous tissue formation, all of which compromise implant stability and contribute to aseptic loosening—a leading cause of implant failure in arthroplasty [6,9]. In total joint replacements, such complications often result in chronic pain, reduced mobility, and the need for costly revision surgeries, significantly impacting patient quality of life and healthcare resources [9,10].

Cooling techniques, such as precooling cement components and saline irrigation, have been explored to mitigate these high temperatures, but their impact on mechanical properties, particularly when combined with different mixing methods, remains underexplored [11,12,13]. Hand mixing, often used in resource-limited settings, introduces higher porosity compared to vacuum mixing, the clinical standard per ISO 5833, which minimizes air entrapment and enhances strength [14,15,16]. This pilot study investigates the effects of hand mixing versus vacuum mixing on the polymerization temperature and compressive strength of antibiotic-loaded PMMA bone cement under two cooling strategies—precooling and saline irrigation. The aim is to compare the influence on thermal management and mechanical integrity of these mixing techniques, potentially reducing thermal necrosis risk while maintaining implant stability in orthopedic applications.

Over the past decade, three-dimensional (3D) printing has emerged as a transformative technology in orthopedic surgery. Additive manufacturing enables the creation of patient-specific anatomical models, surgical guides, and custom implants tailored precisely to a patient’s anatomy [17,18].

In particular, 3D-printed metal implants allow for complex geometries including porous lattice structures that support osseointegration and help reduce stress shielding, thereby potentially improving long-term stability [19].

More advanced developments in 3D bioprinting also promise the fabrication of living, bioactive scaffolds composed of cells and biomaterials, advancing regenerative orthopedics [20].

Because of these capabilities, integrating additive manufacturing into orthopedic biomaterials research—such as the bone–cement systems studied in the present work—is increasingly relevant. The optimization of bone cements, their mechanical behavior, and failure mechanisms can benefit from synergies with 3D-printed scaffold design, personalized fixation, and regenerative strategies.

## 2. Materials and Methods

Antibiotic-loaded Simplex bone cement (Stryker, Kalamazoo, MI, USA) was used, as it is a commonly employed formulation in orthopedic procedures. The cement consists of two sterile components:Liquid monomer: A vial containing 20 mL of a clear, flammable liquid with a mildly sharp odor, composed of 19.5 mL methyl methacrylate (monomer), 0.5 mL N,N-dimethyl-p-toluidine (activator), and 1.5 mg hydroquinone (stabilizer).Powder component: A 41 g packet of fine powder comprising 30.0 g methyl methacrylate–styrene copolymer (containing 1.7% benzoyl peroxide as initiator), 6.0 g polymethyl methacrylate, 4.0 g barium sulphate (radiopacifier, USP and EP compliant), and 1.0 g tobramycin sulphate (antibiotic, USP compliant).

All components were stored at room temperature (23 °C) and 60% humidity prior to preparation, except for the precooled group, as described below.

### 2.1. Sample Preparation

Two batches of cement samples were prepared: one using a standardized hand-mixing technique and another using vacuum mixing, standard to ISO 5833. The hand-mixing batch was used to assess baseline variability in porosity and its effects on cooling strategies, as hand mixing is known to introduce more air entrapment compared to vacuum mixing. While this approach replicates conditions sometimes encountered in resource-limited or rapid-preparation scenarios (e.g., custom antibiotic-loaded spacers), it deviates from the clinical standard of vacuum mixing, which reduces porosity and enhances strength in primary arthroplasties. The vacuum-mixing batch was included to evaluate the cooling strategies under conditions more representative of standard clinical protocols, adhering to ISO 5833 guidelines for mixing and preparation.

For the hand-mixing batch, the preparation process was consistent across all samples and performed by the same investigator to minimize variability. The liquid monomer was poured into a sterile stainless-steel mixing bowl containing the powder component. Manual stirring was initiated immediately using a sterile spatula at a rate of approximately 60 revolutions per minute. Mixing continued for 60 s until a homogeneous, dough-like consistency was achieved without sticking to the spatula or bowl, in accordance with the manufacturer’s guidelines for antibiotic Simplex bone cement. The total mixing phase lasted 1 min, followed by a working phase of 3–5 min at 23 °C, during which the cement was transferred to molds. Polymerization and setting occurred over approximately 10–15 min, as monitored by temperature rise and hardening.

For the vacuum-mixing batch, the cement components were mixed using a vacuum mixing system (e.g., Stryker MixeVac III, Stryker, Kalamazoo, MI, USA) at a vacuum level of −0.8 bar for 60 s, following ISO 5833 standards. This method evacuates air during mixing to minimize porosity, resulting in a more uniform cement matrix. The mixing was performed at room temperature (23 °C) unless specified for the precooled group, and the cement was transferred to molds during the working phase (3–5 min), with polymerization monitored similarly to the hand-mixing batch.

After mixing, each batch (hand-mixed and vacuum-mixed) was divided into three groups, each consisting of 20 samples for mechanical testing and 5 additional samples per group for temperature monitoring (discarded post curing due to structural compromise from embedded thermometers). Each sample was cast into cylindrical molds with a diameter of 1.5 cm and a height of 8 cm, constructed from polypropylene. The molds were filled within 2 min of mixing completion to ensure consistency during the dough phase.

After curing for 24 h at 23 °C, the samples were demolded and cut to a uniform height of 3 cm using a precision diamond saw under irrigation to prevent friction heating. Both ends of each cylinder were sanded with 400-grit sandpaper to ensure smooth, parallel surfaces, minimizing stress concentrations during mechanical testing. The final dimensions were verified using a digital caliper, yielding an average diameter of 1.47 ± 0.02 cm and height of 3.00 ± 0.05 cm (Figure 1).

To ensure exclusion of specimens with excessive porosity, voids, or structural imperfections that could bias mechanical testing outcomes, all prepared cement cylinders underwent radiologic evaluation prior to experimentation. Radiographic screening was performed using a digital X-ray system (Siemens Ysio Max digital X-ray system software version VE10J, which includes the SmartOrtho module and DICOM integration features, 65 kV) calibrated for small, high-contrast objects. Each sample was positioned centrally on the detector and imaged in both anteroposterior and lateral orientations to maximize detection of internal voids (Figure 2).

Porosity was identified as discrete radiolucent areas within the cement matrix, while cracks or irregular inclusions were identified as linear or heterogeneous radiolucencies. Samples exhibiting any of the following were excluded from further analysis: macroporosity (pores larger than 1 mm) and structural imperfections (folds or gaps).

### 2.2. Cooling Techniques

Three distinct groups were established in each batch (hand-mixed and vacuum-mixed) to evaluate the impact of cooling techniques on PMMA polymerization.

Control Group: After mixing, the cement was allowed to harden without any cooling intervention, simulating standard intraoperative conditions. Samples were cured at ambient conditions (23 °C, 60% humidity).

Precooled Group: The liquid monomer and powder components were refrigerated at 6 °C for 12 h (overnight) in a calibrated medical-grade refrigerator prior to mixing. The refrigerated components were mixed immediately upon removal to maintain the low temperature, and the resulting cement was cast and cured at 23 °C.

Saline Irrigation Group: After mixing and casting, the cement samples were irrigated with sterile saline cooled to 8 °C, delivered via a continuous drip system at a rate of 10 mL/min for the first 10 min of polymerization. The saline was maintained at 8 °C using a refrigerated bath circulator to ensure consistent cooling during the exothermic reaction.

Each group had five samples prepared by embedding a thermometer inside the sample to record the changes in core temperatures in real time during the polymerization reaction.

The samples were then subjected to a compressive test by placing each sample into a press and applying an increasing force until mechanical failure. After performing the tests and recording the data, we compared each group to see if the different cooling techniques have any impact on the mechanical proprieties of the cement. The press we opted to use was the C092-06 Compression/exural frame produced by MATEST (Appendix A).

### 2.3. Mechanical Testing

Compressive strength was assessed using a C092-06 Compression/Flexural Frame (MATEST, Treviolo, Italy), a hydraulic press designed for material testing. Each cylindrical sample was placed between two steel disks (6 cm diameter; 5 mm thickness) to ensure uniform load distribution. A compressive force was applied at a constant rate of 1 mm/min until mechanical failure, defined as the point of maximum load before fracture or significant deformation. The maximum force (F, in Newtons) was recorded, and compressive strength (σ, in megapascals, MPa) was calculated using the formula:σ = F/A
where A = π * r^2^ (r = radius of the cylinder in meters, π ≈ 3.14159).

ASTM F451 (Standard Specification for Acrylic Bone Cement) and ISO 5833 typically recommend a constant crosshead speed of 0.5 to 1 mm/min for quasi-static compressive testing to ensure consistent and reproducible results. Testing speed was in accordance with this, specifically 1 mm/min, as this is common for evaluating mechanical properties of bone cement under controlled conditions.

The method for measuring displacement was the linear variable differential transformer (LVDT) built into the MATEST C092-06 frame. This allows precise measurement of the crosshead displacement during compression.

Displacement was recorded continuously by the machine’s data acquisition system, which monitors the movement of the loading platen as force is applied. The system logs displacement in millimeters, correlated with the applied force (in kN), to calculate compressive strength using the formula provided, σ = F/A, where F is the maximum force at failure (in Newtons) and A is the cross-sectional area of the cylinder (A = π * r^2^, with r = 0.735 cm based on the 1.47 cm diameter).

The failure point was determined when the sample exhibited mechanical failure, such as cracking or shear failure, as observed in the visual inspection of deformed cylinders. The maximum force at failure was used to compute compressive strength in megapascals (MPa).

Post testing, the failure patterns of the samples were visually inspected and categorized (e.g., uniform cracking, shear failure, crumbling) to assess differences in mechanical behavior across groups.

### 2.4. Temperature Monitoring

For each batch, core temperature was monitored in real-time during polymerization using the embedded PT100 thermocouple connected to a digital data logger (accuracy ± 0.1 °C). The sensor was positioned at the geometric center of the cylindrical mold to capture peak temperatures accurately. No additional additives, beyond the silicone grease for sensor removal, were applied to the temperature-monitoring samples. Temperature data were recorded in real-time at 10 s intervals for 18 min, covering the full polymerization period. These samples were discarded after curing due to potential structural weakening from the thermometer insertion. Peak temperatures and temperature profiles were analyzed to compare thermal behavior across the control, precooled, and saline irrigation groups.

### 2.5. Statistical Analysis

Compressive strength data were analyzed using Welch’s *t*-test to compare the precooled and saline irrigation groups against the control group, accounting for potential unequal variances. Outliers, defined as values beyond 1.5 times the interquartile range, were identified and excluded to ensure robust statistical comparisons. A significance threshold of *p* < 0.05 was adopted. No correction for multiple comparisons was applied, as only two pairwise tests (precooled vs. control and saline irrigation vs. control) were performed. However, this increases the risk of Type I error inflation, and the results should be interpreted with caution, particularly for the saline irrigation group, where the effect size is small. Descriptive statistics, including mean, standard deviation (SD), range, and 95% confidence intervals, were calculated for each group. Statistical analyses were performed using GraphPad Prism version 9.0 (GraphPad Software, San Diego, CA, USA).

## 3. Results

All reported mechanical values represent compressive strength (MPa), calculated from the maximum load at failure divided by specimen cross-sectional area.

Table 1 presents the compressive strength results for the 20 samples per group. The control group exhibited an average compressive strength of 14.59 MPa, with values ranging from 13.2700 to 15.5160 MPa. The precooled group showed an average compressive strength of 12.45 MPa, with values ranging from 10.4620 to 15.2810 MPa. The saline irrigation group had an average compressive strength of 14.64 MPa, with values ranging from 13.3880 to 15.8690 MPa (Table 1, Figure 3).

Core temperature measurements during polymerization showed that the control group reached peak temperatures of 75–77 °C. The precooled group, with components initially at 6 °C, exhibited peak temperatures of 63–65 °C. The saline irrigation group, cooled with 8 °C saline, maintained peak temperatures of 65–67 °C. (Appendix A, Figure 4).

Visual inspection of the deformed cylinders revealed that the control group predominantly exhibited uniform cracking (70% of samples, *n* = 14). The precooled group showed a higher incidence of shear failure (50% of samples, *n* = 10). The saline irrigation group displayed a mix of cracking (60%, *n* = 12) and minor crumbling (30%, *n* = 6).

The statistical analysis in the manuscript utilized Welch’s *t*-test to compare the mean compressive strengths between the experimental groups (pre-cooled and saline irrigation) and the control group for the two separate batches of cement. This non-parametric variant of the independent samples *t*-test was selected due to the unequal variances observed across groups, which makes it robust to violations of the equal variance assumption without requiring data transformation (Table 2).

Normality of the distributions of the vacuum-mixed batch was assessed using the Shapiro–Wilk test (*p* > 0.05 for control and saline-irrigated groups; *p* = 0.075 for pre-cooled group, confirming approximate normality). Homogeneity of variances was evaluated with Levene’s test (*p* > 0.05 for both comparisons, but Welch’s *t*-test was used conservatively for robustness).

A two-tailed significance level (α) of 0.05 was applied, consistent with standard conventions in biomedical research for Type I error control. Effect sizes were calculated using Cohen’s d (1.52 for pre-cooled vs. control, indicating a large effect; 0.85 for saline irrigation vs. control, indicating a large effect). Post hoc power analysis confirmed adequate statistical power (>0.80) for detecting the observed differences based on the sample size (*n* = 20 per group after exclusions) for the pre-cooled comparison (power = 0.997), and borderline adequate power for the saline irrigation comparison (power = 0.749). No multiple comparison corrections (e.g., Bonferroni) were applied, as only pairwise comparisons to the control were performed.

Normality of the distributions in the hand-mixed batch was assessed using the Shapiro–Wilk test (*p* > 0.05 for control and saline-irrigated groups; *p* = 0.027 for pre-cooled group, indicating mild deviation from normality but proceeding with parametric tests for consistency with the protocol). Homogeneity of variances was evaluated with Levene’s test (*p* > 0.05 for both comparisons, but Welch’s *t*-test was used conservatively for robustness).

A two-tailed significance level (α) of 0.05 was applied, consistent with standard conventions in biomedical research for Type I error control. Effect sizes were calculated using Cohen’s d (0.64 for pre-cooled vs. control, indicating a medium effect; −0.07 for saline irrigation vs. control, indicating negligible effect). Post hoc power analysis confirmed borderline statistical power (0.50) for detecting the observed differences based on the sample size (*n* = 20 per group after exclusions) for the pre-cooled comparison, and inadequate power for the saline irrigation comparison (power = 0.056). No multiple comparison corrections (e.g., Bonferroni) were applied, as only pairwise comparisons to the control were performed (Table 2).

## 4. Discussion

Polymethyl methacrylate (PMMA) bone cement remains essential in orthopedic surgery for prosthetic fixation, but its exothermic polymerization can exceed 80 °C, risking thermal osteonecrosis at the bone–cement interface [21]. This pilot study evaluated the impact of two cooling strategies—precooling components and saline irrigation—on polymerization temperature and compressive strength, comparing hand-mixing (HM) and vacuum-mixing methods using antibiotic-loaded Simplex cement.

### 4.1. Effects of Cooling on PMMA Bone Cement

Both cooling techniques effectively reduced peak polymerization temperatures below those of controls, with precooling showing the greatest mitigation. For HM, controls peaked at 76.2 °C (15 min), while precooled samples reached 63.6 °C and saline-irrigated 66 °C. Vacuum-mixing controls peaked at 71.8 °C, with precooled at 58.8 °C and saline at 63.6 °C (Appendix A). These reductions align with prior work, such as Tai et al., who reported 10–15 °C drops with precooling to 3 °C, extending handling time by up to 50% [22]. The slower reaction kinetics at lower initial temperatures delay viscosity rise, improving surgical precision [8,23]. However, saline irrigation’s modest effect (5–8 °C reduction) remains above the 50 °C necrosis threshold, limiting its standalone efficacy against thermal damage [23,24].

### 4.2. Mechanical Properties After Cooling

Compressive strength varied markedly by mixing method, with samples prepared using vacuum mixing achieving 76–80 MPa (comparable to ISO 5833 standards) versus 16–17 MPa for HM, reflecting porosity differences from air entrapment in hand mixing [25]. Within HM, precooling reduced mean strength to 16.49 MPa (SD: 0.90) from 17.02 MPa in controls (*p* = 0.051, Cohen’s d = 0.64; Table 2), with higher variability indicating inconsistent polymer network formation. Saline irrigation yielded 17.07 MPa (*p* = 0.820, negligible effect), preserving integrity. In vacuum-mixed samples, precooling lowered strength to 78.45 MPa from 79.73 MPa (*p* = 0.000027, d = 1.52; Table 2), while saline reached 76.77 MPa (*p* = 0.010, d = 0.85), both retaining high values suitable for load-bearing.

These trends contrast with some studies, where precooling to 3–6 °C preserved strength under vacuum mixing [26,27]. The observed reductions likely stem from study-specific factors: hand-mixing-induced porosity, antibiotic loading (increasing voids [28]), larger specimens (15 × 30 mm vs. ISO 6 × 12 mm), and 24 h curing (though residual monomer effects possible). Failure modes supported this—controls showed uniform cracking, while precooled exhibited more shear failure, suggesting flaw sensitivity. Saline irrigation’s neutral impact on HM makes it preferable for resource-limited settings using hand mixing.

#### Differences in Failure Mechanism

Changes in the cooling conditions altered not only the compressive strength but also the dominant failure mechanisms observed. In the control groups, most specimens exhibited uniform, circumferential cracking typical of a relatively homogeneous PMMA microstructure.

Experimental studies demonstrate that lowering the temperature (pre-cooling) increases the yield stress and shear strength of PMMA, making it more resistant to shear failure. Specifically, mechanical testing at sub-ambient temperatures (e.g., 0 °C or −40 °C) shows that PMMA becomes harder and stiffer, with increased resistance to deformation and failure under shear and compressive loads [29,30]. The molecular basis is that reduced temperature limits polymer chain mobility, increasing the material’s brittleness but also its strength against shear forces.

Shear failure in PMMA is more likely at elevated temperatures, where the polymer transitions toward a more ductile, viscoelastic state and the shear strength decreases [29,31,32]. At lower temperatures, PMMA exhibits higher yield stress and modulus, and the incidence of shear failure is reduced unless the material is subjected to extremely high strain rates or impact loading.

When saline-irrigated polymethyl methacrylate (PMMA) samples are tested for compressive strength, the observed mechanical failure modes include mixed cracking—characterized by the formation of cracks propagating through the matrix and around undissolved PMMA beads and minor surface crumbling, which manifests as localized surface fragmentation and small-scale material loss at the compression interface [33]. Additionally, compressive failure is associated with the development of a “yielded crack band” across the transverse section, indicating plastic deformation and squashing of PMMA beads in the longitudinal direction [33]. The presence of saline can alter the microstructure, potentially increasing porosity and promoting microcrack initiation at pore perimeters, which may further contribute to non-linear damage evolution and surface crumbling under load [34,35].

Overall, the predominant failure types in saline-irrigated PMMA under compression are mixed cracking and minor surface crumbling, with microcrack accumulation and plastic deformation also contributing to the loss of mechanical integrity [33,34,35].

### 4.3. Formulation and Procedural Considerations

Antibiotic-loaded cements inherently increase porosity, exacerbating cooling’s effects on strength [36,37]. Vacuum mixing mitigated this, aligning results with clinical benchmarks. Saline exposure during irrigation minimally disrupted polymerization in HM, unlike reports of moisture-induced weakening [28,34,38], possibly due to brief application (10 min).

### 4.4. Clinical Implications

Precooling offers superior thermal control but risks strength compromise in non-vacuum scenarios, suitable for vertebroplasty needing extended handling [38] but cautious in arthroplasties. Saline irrigation balances modest cooling with preserved mechanics in HM, potentially reducing necrosis risk intraoperatively without added equipment. Vacuum mixing remains the standard for primary procedures, where cooling effects are minor. Combining strategies (e.g., vacuum with saline) warrants exploration for optimized fixation [39,40,41,42].

Our findings on compressive strength and failure mechanisms have important implications for patient-specific orthopedic applications, especially in the era of additive manufacturing. As 3D printing becomes more integrated into orthopedic implant design, understanding how materials like bone cement perform under different thermal and mechanical conditions is critical. For instance, custom 3D-printed implants increasingly leverage lattice architectures to promote osseointegration and reduce mismatch with bone mechanical properties [17,18].

Moreover, the rise of bioprinted scaffolds in orthopedics combining living cells, polymers, and bioceramics underscores the need for materials that maintain structural integrity and bioactivity under complex loading and temperature variations [19].

By characterizing how cooling methods (pre-cooling, saline irrigation) affect cement polymerization, strength, and failure behavior, our study contributes foundational data that may guide the design of hybrid constructs: for example, 3D-printed implants or scaffolds that are anchored or supplemented with optimized bone cement. Ultimately, such integration may enhance the performance and longevity of personalized orthopedic implants [20].

### 4.5. Implications for Drug-Containing Bone Cements

Given the growing clinical importance of local drug delivery in orthopedics (e.g., antibiotic-loaded bone cement in periprosthetic joint infection), our mechanical and polymerization findings acquire additional relevance. Traditional PMMA-based antibiotic-loaded bone cements (ALBCs) remain the clinical standard because they combine high mechanical strength and injectability. However, simply mixing high doses of drug into PMMA often leads to drawbacks: much of the drug may remain trapped within the polymer matrix, burst-release profiles, and compromised mechanical integrity [43].

To overcome these limitations, advanced production methods may be more suitable:Use of porogens or nanostructured additives (e.g., nanocellulose, halloysite nanotubes) can increase cement porosity or serve as hydrophilic pathways, thereby enhancing elution without excessively weakening the material [43].Incorporation of carrier systems within the cement—such as liposomes loaded with antibiotic—can protect the drug during cement polymerization and allow more controlled release [44].Embedding biodegradable microparticles that contain the therapeutic agent into a cement matrix can provide a sustained, gradual release as the particles degrade [45].Alternative cement matrices, such as calcium phosphate-based cements, may offer better biocompatibility and resorbability, making them favorable for long-term drug delivery, though they may not provide the same mechanical robustness as PMMA [46].

Depending on the clinical application, different strategies may be optimal. For instance, in infection treatment settings, a high initial antibiotic release (burst) followed by prolonged elution may be desired, favoring additive- or carrier-modified PMMA. In regenerative applications, biodegradable cements delivering growth factors or small molecules might be more appropriate [43,44,45].

Our study’s observations on how cooling affects polymerization kinetics and mechanical strength may guide the formulation of drug-loaded cements, especially when using sensitive bioactive agents: controlling exotherm and polymer structure could help preserve drug activity and release behavior.

### 4.6. Innovation and Importance of This Research

Although PMMA bone cement has been used in orthopedics for decades, the combined influence of mixing technique and cooling method on both polymerization temperature and mechanical performance has not been thoroughly characterized. Previous studies have evaluated individual factors such as precooling alone, antibiotic loading, or vacuum mixing but no prior work has directly compared hand mixing versus vacuum mixing under multiple clinically relevant cooling strategies using antibiotic-loaded cement. This integrated approach represents a key innovation of the present study [27,38].

From a clinical perspective, thermal necrosis at the bone–cement interface remains a persistent concern, especially in procedures with large cement volumes or confined anatomical spaces. At the same time, modifications that reduce polymerization temperature often compromise cement strength. By systematically evaluating thermal behavior, compressive strength, and failure mechanisms under different cooling conditions, our study provides new insights that help balance these competing priorities [38].

The findings are particularly important because they reflect real-world surgical variability: hand mixing is still widely used in resource-limited settings or during rapid intraoperative preparation (e.g., spacers for infection), while vacuum mixing represents the modern ISO standard for arthroplasty. Demonstrating how cooling strategies interact with these mixing methods helps clarify which techniques are safe, which may jeopardize implant stability, and which offer the best thermal control without sacrificing mechanical integrity.

Moreover, as orthopedics advances toward personalized implants, additive manufacturing, and drug-releasing biomaterials, understanding how cement polymerization can be modulated is increasingly relevant. The ability to control temperature without degrading material performance may directly support the development of patient-specific constructs, hybrid 3D-printed–cement interfaces, and optimized antibiotic-loaded cements. Thus, the present study contributes foundational knowledge that informs both current surgical practice and the evolving landscape of orthopedic biomaterials design [18].

### 4.7. Additional Material Characterization Considerations

Although radiographic screening was used to exclude samples with macroporosity and major defects, more advanced characterization techniques including quantitative porosity analysis, scanning electron microscopy (SEM), and Fourier transform infrared spectroscopy (FTIR) would provide deeper insight into the microstructural and chemical mechanisms underlying the differences observed between cooling strategies [47,48].

Porosity plays a critical role in PMMA mechanical performance. Prior work has shown that air-cured PMMA bone cement may exhibit porosity levels up to 16.8%, whereas vacuum-treated samples demonstrate substantially reduced porosity around 5.1%. Additives commonly used in clinical and research contexts such as antibiotics, porogens, or bioactive fillers can further increase micropore density (0.1–0.5 mm) and total pore volume, resulting in changes to strength, crack propagation behavior, and elution kinetics. These effects are particularly relevant to our findings, as hand mixing inherently introduces higher porosity, amplifying the influence of cooling on failure modes [48,49].

SEM imaging in previous studies (including our own) has revealed that air-cured PMMA typically displays irregular voids, aggregated PMMA beads, and visible radiopaque particles. In contrast, vacuum-treated or porogen-modified formulations show more homogeneous and interconnected pore networks, with microstructural patterns strongly influenced by the additives used. Porosity-modifying agents such as carboxymethylcellulose or effervescent compounds can generate open pore architectures that support osteoblast infiltration and bone ingrowth features that would be relevant for future bioactive or regenerative cement formulations [50,51,52]

FTIR spectroscopy consistently identifies characteristic PMMA absorption peaks, such as the ester carbonyl stretch (≈1720 cm^−1^), C–O stretching (≈1140 cm^−1^), and C–H stretching modes (≈2950 cm^−1^). The incorporation of additives like calcium phosphate, hydroxyapatite, or boron-containing fillers does not alter the fundamental PMMA chemical backbone, but introduces additional bands corresponding to the functional groups of the additives, confirming their chemical integration [53].

Together, these characterization modalities, including porosity measurement, SEM morphology, and FTIR chemical analysis, provide complementary insights into the tunable nature of PMMA bone cement. They could directly support interpretation of cooling-induced changes in polymer architecture and mechanical behavior in future work [54,55].

### 4.8. Limitations and Future Directions

A major limitation of this pilot study is the absence of advanced material characterization methods such as scanning electron microscopy (SEM), Fourier transform infrared spectroscopy (FTIR), and quantitative porosity analysis. These techniques would provide deeper insight into pore morphology, inter-bead fusion, microcracking, and potential chemical or polymerization-related changes induced by cooling. Although beyond the scope of the present pilot design, such analyses are indispensable for fully evaluating the basis of the observed mechanical and thermal outcomes and will be included in future work. The present findings should therefore be interpreted as functional thermo-mechanical trends rather than microstructural explanations

Second, testing was limited to compressive loading, whereas clinical bone cement is subjected to complex multi-axial stresses, including tension, bending, torsion, fatigue loading, and interfacial shear at the cement–bone interface. Expanding evaluation to include fatigue testing, fracture toughness, and long-term mechanical stability would strengthen the translational relevance of our findings.

Third, this study examined only one commercially available antibiotic-loaded PMMA formulation, one antibiotic type, and two cooling techniques. Alternative formulations including high-dose antibiotic cements, additive-modified cements, porous cements, or bioactive/biodegradable composites may respond differently to thermal modulation. Likewise, additional cooling approaches (e.g., pulsed irrigation, cooled applicators, pre-cooled prosthesis surfaces) warrant investigation.

Fourth, the laboratory environment cannot fully replicate the clinical setting, where cement is applied to irregular bone surfaces, exposed to body temperature, blood, marrow fat, and fluctuating intraoperative conditions. In vivo thermal behavior may differ substantially from benchtop measurements. Animal models or computational simulations could help clarify thermal transfer to surrounding tissues and the true risk of thermal osteonecrosis.

Finally, as additive manufacturing and drug delivery applications continue to expand in orthopedics, the interaction between cooling strategies and advanced cement formulations will require dedicated study. Future research should explore how thermal modulation influences antibiotic elution kinetics, integration with 3D-printed implants, and the performance of hybrid constructs such as cement-augmented, porous-lattice prostheses.

Overall, this study provides an initial framework for understanding how mixing and cooling interact to shape the thermal and mechanical behavior of PMMA bone cement, but further multi-modal and clinically oriented investigations are needed to optimize cement performance in modern orthopedic applications.

## 5. Conclusions

This pilot study evaluated the effects of precooling and saline irrigation on the polymerization temperature and compressive strength of antibiotic-loaded PMMA bone cement, comparing hand mixing and vacuum mixing. Hand-mixed samples exhibited significantly lower compressive strengths (16–17 MPa) compared to vacuum-mixed (76–80 MPa), reflecting increased porosity from air entrapment during hand mixing, which deviates from the ISO 5833 vacuum-mixing standard. This porosity amplified the impact of cooling, particularly with precooling, which reduced hand-mixed sample strength to 16.49 MPa (*p* = 0.051) from 17.02 MPa in controls, and vacuum-mixed sample strength to 78.45 MPa (*p* < 0.001) from 79.73 MPa. Saline irrigation maintained mechanical integrity in HM (17.07 MPa, *p* = 0.820) and showed minimal reduction in vacuum-mixed samples (76.77 MPa, *p* = 0.010), suggesting relative safety for both cooling techniques when combined with proper preparation.

Precooling significantly lowered peak temperatures (hand-mixed: 63.6 °C and vacuum-mixed: 58.8 °C vs. controls at 76.2 °C and 71.8 °C), enhancing handling time but risking strength compromise in hand-mixed scenarios. Saline irrigation provided moderate thermal control (hand-mixed: 66 °C; vacuum-mixed: 63.6 °C) while preserving mechanical properties, offering a practical intraoperative option, particularly for hand mixing in resource-limited settings. Vacuum mixing, as the clinical standard, consistently yielded superior mechanical performance, aligning with ISO benchmarks and mitigating cooling-related strength reductions.

These findings highlight the critical role of mixing technique in PMMA performance, with vacuum mixing ensuring robust mechanical properties suitable for load-bearing applications. Both cooling strategies are safe under vacuum mixing, maintaining sufficient strength, but hand-mixed cement requires cautious application due to inherent porosity.

## Figures and Tables

**Figure 1 biomedicines-13-03071-f001:**
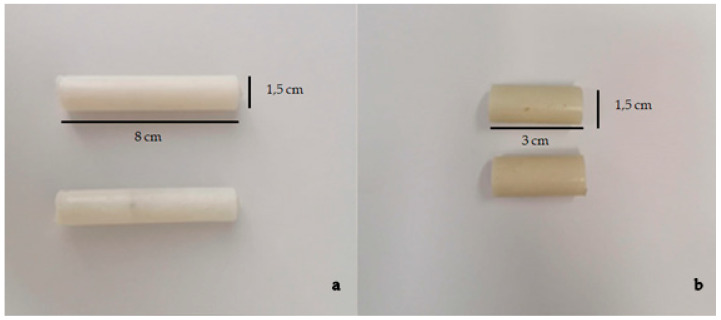
(**a**) The cement cylinders after curing (8 × 1.5 cm) and (**b**) after curing and preparation (3 × 1.5 cm) after preparation.

**Figure 2 biomedicines-13-03071-f002:**
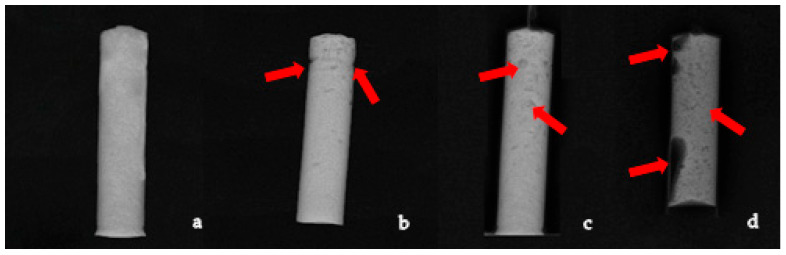
Images of radiographic control of cement samples, red arrows are used to indicate porosity in samples b and c, while they show catastrophic structural defects and porosity in sample d. (**a**) Example of accepted cement samples; (**b**–**d**) example of rejected samples; (**b**) a structural defect visible at the top right part and increased porosity at the top third of the sample; (**c**) example of a sample that exhibited no signs of structural defects, but was excluded after radiographic control, due to increased porosity at the top half of the sample; (**d**) example of a sample that exhibited catastrophic structural failure on the right side and unacceptable levels of porosity throughout the body of the cylinder.

**Figure 3 biomedicines-13-03071-f003:**
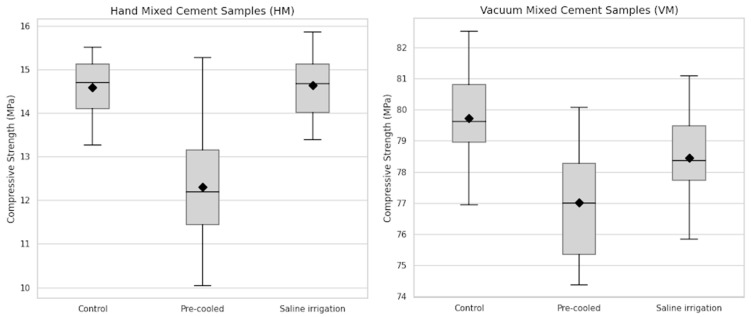
Compressive strength values for each of the HM cement batches: the control batch with average compressive strength value of 14.59 MPa, SD: 0.64 MPa; the pre-cooled batch with a mean compressive strength of 12.45 MPa, SD: 1.22 MPa; and the saline-irrigated batch with a mean compressive strength of 14.64, SD: 0.66 MPa, respectively. Compressive strength values for the VM cement batches: the control batch with average compressive strength value of 79.73 MPa, SD: 1.54 MPa; the pre-cooled batch with a mean compressive strength of 77.02 MPa, SD: 1.99 MPa; and the saline-irrigated batch with a mean compressive strength of 78.45, SD: 1.44 MPa.

**Figure 4 biomedicines-13-03071-f004:**
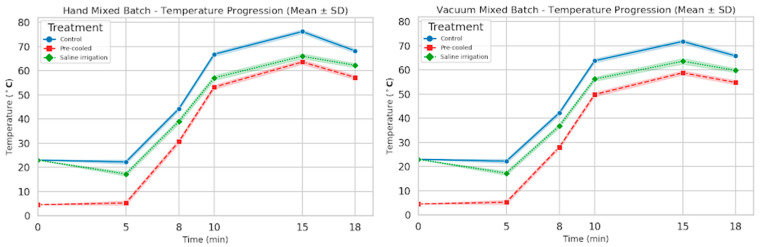
Temperature curve of each cement sample from the three different batches for the two groups (vacuum-mixed and hand-mixed, respectively); blue—control; red—pre-cooled; green—saline irrigation.

**Table 1 biomedicines-13-03071-t001:** The average compressive strength of each batch of cement cylinders (HM—hand-mixed; VM—vacuum-mixed).

Sample Group	Average Compressive Strength Values of Samples (MPa)
HM control	17.0147 ± 0.64
HM pre-cooled	16.48735 ± 1.22
HM saline irrigation	17.06975 ± 0.66
VM control	79.726 ± 1.54
VM pre-cooled	78.454 ± 1.99
VM saline irrigation	76.770915 ± 1.44

**Table 2 biomedicines-13-03071-t002:** Statistical values for the hand-mixed PMMA bone cement.

Comparison	t-Statistic	*p*-Value	Cohen’s d	Power
Pre-cooled HM vs. control HM	2.01	0.051	0.64	0.501
Saline HM vs. control HM	−0.23	0.820	−0.07	0.056
Pre-cooled VM vs. control VM	4.81	0.000027	1.52	0.997
Saline VM vs. control VM	2.70	0.010	0.85	0.749

## Data Availability

The raw data supporting the conclusions of this article will be made available by the authors on request.

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
