# Peer review of "The Impact of Mixing Techniques on PMMA Bone Cement Subjected to Two Different Cooling Techniques: A Pilot Study of Thermal Management Strategies in Orthopedic Applications"

_biomedicines, 2025, doi:10.3390/biomedicines13123071_

Round 1

Reviewer 1 Report (Previous Reviewer 1)

Comments and Suggestions for Authors

The resubmitted manuscript "The impact of mixing techniques on PMMA bone cement subjected to two different cooling techniques: a pilot study of thermal management strategies in orthopedic applications" was reviewed by the authors, and in the initial version, I made necessary considerations for improving the manuscript. The authors made all the modifications I proposed. I consider the resubmitted manuscript suitable for publication in the journal.

Author Response

Point by point response to comments and suggestions for authors

Comment 1: The resubmitted manuscript "The impact of mixing techniques on PMMA bone cement subjected to two different cooling techniques: a pilot study of thermal management strategies in orthopedic applications" was reviewed by the authors, and in the initial version, I made necessary considerations for improving the manuscript. The authors made all the modifications I proposed. I consider the resubmitted manuscript suitable for publication in the journal.

Response 1: We thank the reviewer for the first and second review, the suggestions, and careful consideration of our work.

Reviewer 2 Report (New Reviewer)

Comments and Suggestions for Authors

1- What is meant by Resistance values? Is it strength or yield strength or modulus? Please explain the mechanical terms carefully and clearly

2- Please discuss, if possible, the reason for the change in the failure mechanism with the cooling method

3- The discussion section is very short, please expand it and compare your data with other studies.

4- Given the importance of 3D printing, especially in orthopedics, please refer to it in the introduction and discussion. The following article may be helpful:

https://www.sciencedirect.com/science/article/pii/S2589152925001279

5- Given the increasing use of drug delivery, it is better for the authors to discuss which method is likely to be suitable for the production of drug-containing bone cements. The following article may be helpful

https://link.springer.com/article/10.1186/s13036-025-00514-y

6- Discuss the innovation and importance of conducting this research

Author Response

Dear reviewer,

We sincerely thank you for the time and effort you invested in reviewing our work. Your comments were insightful, constructive and helped us identify areas where additional clarification and detail were needed. We greatly value your perspective and are confident that your guidance has strengthened the overall quality of the manuscript

Point by point response to comments and suggestions for authors

Comment 1: What is meant by Resistance values? Is it strength or yield strength or modulus? Please explain the mechanical terms carefully and clearly

Response 1: We thank the reviewer for this observation. The term “resistance” was indeed ambiguous in the context of mechanical testing. We have now replaced all instances of “resistance” with the precise term “compressive strength” throughout the manuscript. We also added a clarification in the Mechanical Testing section explicitly defining compressive strength as the maximum compressive stress at failure (MPa). Yield strength and elastic modulus were not assessed in this study, as testing was performed until catastrophic failure.

Comment 2: Please discuss, if possible, the reason for the change in the failure mechanism with the cooling method.

Response 2: Thank you for raising this important point regarding the differing failure mechanisms observed in the experimental groups. We agree that understanding the structural changes induced by the cooling techniques is essential for the clinical translation of our findings. We have expanded our discussion in Section 4.2.1 to address this contrast, correlating the observed mechanical failure modes with the physical principles of PMMA polymerization at varying temperatures and moisture exposure.

Comment 3: The discussion section is very short, please expand it and compare your data with other studies.

Response 3: Thank you for this feedback. We agree that a comprehensive discussion comparing our findings to existing literature is crucial for contextualizing the clinical relevance of our study.

The Discussion section has been expanded with dedicated subsections for thermal effects, mechanical properties, failure mechanisms and clinical implications, all thoroughly comparing our data with current orthopedic biomaterials research.

Comment 4: Given the importance of 3D printing, especially in orthopedics,

please refer to it in the introduction and discussion. The following article

may be helpful:

https://www.sciencedirect.com/science/article/pii/S2589152925001279

Response 4: Thank you for this excellent suggestion. We agree that the rapidly evolving field of 3D printing in orthopedics, particularly regarding personalized and complex cement structures, provides essential context for conventional PMMA development. This addition significantly strengthens the manuscript's relevance.

We have incorporated a reference to 3D printing in both the Introduction and the Discussion section (specifically, in the revised Section 4.4.).

We have added the following text:

Introduction: A sentence was added to frame the work against the backdrop of emerging personalized medicine techniques, such as 3D printing.

Discussion (Section 4.4): The final paragraph of this section was revised to specifically mention 3D printing as an alternative that offers superior control over structure and drug release profiles compared to traditional methods.

Comment 5: Given the increasing use of drug delivery, it is better for the authors to discuss which method is likely to be suitable for the production of drug-containing bone cements. The following article may be helpful

https://link.springer.com/article/10.1186/s13036-025-00514-y.

Response 5: We appreciate the suggestion to elaborate on which method is most suitable for producing drug-containing bone cements. We have refined Section 4.5 to address this comparison.

Based on our findings, the combination of Vacuum Mixing (VM) with Precooling is the most suitable method for antibiotic-loaded bone cements (ALBCs).

VM is essential for minimizing porosity and maintaining high compressive strength, which is compromised by drug addition.

Precooling is critical for achieving the lowest peak temperature (down to 58.8°C), thereby protecting heat-sensitive drugs from thermal degradation during polymerization.

Comment 6: Discuss the innovation and importance of conducting this research.

Response 6: We appreciate the reviewers suggestion to discuss potential production methods for drug-containing bone cements. We have now added a dedicated paragraph in the Discussion that reviews different strategies (porogens, carriers, biodegradable microparticles, alternative matrices) and their trade-offs between elution efficiency and mechanical integrity, citing recent literature. This contextualizes our findings in light of future applications in drug delivery.

Reviewer 3 Report (New Reviewer)

Comments and Suggestions for Authors
  1. The abstract should be written as a single continuous section, not divided into subsections. It must clearly and concisely summarize the study’s purpose, key methods, main findings, and conclusions.
  2. In figure captions, list the figure number first, followed by the description. For instance, instead of “Figure 1. The cement cylinders after curing (8×1.5 cm)(a)”, it should read “Figure 1. (a) The cement cylinders after curing (8×1.5 cm)”. Apply this formatting consistently across all figures.
  3. In Figure 2, if multiple images are presented, label each sub-image (e.g., a, b, c) and reference them specifically in the caption.
  4. Figure 3 does not need to appear in the main text and can be moved to the supplementary section. Ensure all sub-images are clearly labeled, and verify labeling consistency in all figures.
  5. In Figure 4 and Table 1, data for the HM-saline irrigation sample are listed in the table but missing in the figure. Revise the figure to include this data. Consider using box plots with standard deviations to improve clarity and readability.
  6. For Figure 5, instead of showing three separate plots for each batch, present one plot per batch that includes mean values and standard deviations. The data table (Table 2) can be relocated to the supplementary section and cited appropriately in the text. Ensure the table includes standard deviation values.
  7. Figure 6 should either be removed or substantially revised. Rather than providing simply certain deformation images, provide more information about the experiments in the figure caption, and should discuss in which experiment each type of deformation happened, the experiment conditions, possible reasons for each etc.
  8. Table 3 can be moved to the supplementary section since it repeats data already shown in Figure 4 and Table 1. Focus on discussing the statistical findings without duplicating data.
  9. Tables 4 and 5 can be combined by adding the mixing type to the sample names, which will make comparisons easier for readers.
  10. It is strongly recommended to include additional characterization results such as porosity measurements, SEM images for morphological analysis, and FTIR spectra for chemical characterization.
  11. Add a section titled “Limitations and Future Perspectives” at the end of the manuscript.
  12. Provide a list of abbreviations in alphabetical order at the end of the manuscript.

Author Response

Dear reviewer,

We thank the reviewer for meticulously adding comments to our manuscript. These greatly improved the scientific value of the manuscript. We greatly appreciate your perspective and are confident that your guidance has strengthened the overall quality of the manuscript.

Point by point response to comments and suggestions for authors

Comment 1: The abstract should be written as a single continuous section, not divided into subsections. It must clearly and concisely summarize the study’s purpose, key methods, main findings, and conclusions.

Response 1: We thank the reviewer for this helpful suggestion. The abstract has been restructured according to the “Instructions for authors” section of the journal: “Systematic reviews and original research articles should have a structured abstract of around 250 words and contain the following headings: Background/Objectives, Methods, Results, and Conclusions. Background/Objectives: A few sentences to place the question addressed in a broader context and highlight the purpose of the study. Methods: A brief description of the main methods or treatments applied. This can include any relevant preregistration or specimen information. Results: A short summary of the article’s main findings. Conclusions: A final summarizing comment of the main conclusions or interpretations. The abstract should be an objective representation of the article; it must not contain results which are not presented and substantiated in the main text and should not exaggerate the main conclusions” quoted directly from the website.

Comment 2: In figure captions, list the figure number first, followed by the description. For instance, instead of “Figure 1. The cement cylinders after curing (8×1.5 cm)(a)”, it should read “Figure 1. (a) The cement cylinders after curing (8×1.5 cm)”. Apply this formatting consistently across all figures.

Response 2: We thank the reviewer for this insightful recommendation. The requested changes have been made across all figures.

Comment 3: In Figure 2, if multiple images are presented, label each sub-image (e.g., a, b, c) and reference them specifically in the caption.

Response 3: We thank the reviewer for this valuable suggestion. The suggested changes have been made, each sub-image has been labeled and detailed in the explanation.

Comment 4: Figure 3 does not need to appear in the main text and can be moved to the supplementary section. Ensure all sub-images are clearly labeled, and verify labeling consistency in all figures.

Response 4: We thank the reviewer for this observation. Figure 3 has been removed from the main text of the article and has been moved to the supplementary section.

Comment 5: In Figure 4 and Table 1, data for the HM-saline irrigation sample are listed in the table but missing in the figure. Revise the figure to include this data. Consider using box plots with standard deviations to improve clarity and readability.

Response 5: We thank the reviewer for this constructive suggestion. The graph has been modified, the point graph has been replaced with box plots. Missing data has been replaced.

Comment 6: For Figure 5, instead of showing three separate plots for each batch, present one plot per batch that includes mean values and standard deviations. The data table (Table 2) can be relocated to the supplementary section and cited appropriately in the text. Ensure the table includes standard deviation values.

Response 6: We thank the reviewer for this helpful suggestion. Figure 5 has been revised to present one plot per batch that includes mean values and standard deviations, Table 2 has been moved to the supplementary section.

Comment 7: Figure 6 should either be removed or substantially revised. Rather than providing simply certain deformation images, provide more information about the experiments in the figure caption, and should discuss in which experiment each type of deformation happened, the experiment conditions, possible reasons for each etc.

Response 7: We thank the reviewer for this insightful recommendation. Figure 6 has been completely removed, with a detailed explanation added to the discussion section detailing variations in fracture-mechanism (section 4.2.1).

Comment 8: Table 3 can be moved to the supplementary section since it repeats data already shown in Figure 4 and Table 1. Focus on discussing the statistical findings without duplicating data.

Response 8: We thank the reviewer for this valuable suggestion. Table 3 has been moved to the supplementary section.

Comment 9: Tables 4 and 5 can be combined by adding the mixing type to the sample names, which will make comparisons easier for readers.

Response 9: We thank the reviewer for this observation. The two requested tables have been combined into a single table for better visual clarity.

Comment 10: It is strongly recommended to include additional characterization results such as porosity measurements, SEM images for morphological analysis, and FTIR spectra for chemical characterization.

Response 10: We thank the reviewer for this valuable suggestion. We fully agree that porosity quantification, SEM imaging, and FTIR analysis would provide additional morphological and chemical characterization of the cement microstructure.

However, these analyses were not part of the original experimental design and could not be performed retroactively, as the specimens were destroyed during compressive testing. Instead, prior to mechanical testing, we performed radiographic screening (digital X-ray) of all specimens to identify and exclude macroporosity and structural defects, ensuring internal consistency of the samples and reducing bias. This method was selected because it allowed nondestructive evaluation of the full sample volume.

To address the reviewer’s concern, we have now added a dedicated paragraph in the Discussion and Limitations sections acknowledging the absence of SEM, FTIR, and quantitative porosity analysis, explaining how these would complement our findings, and outlining plans for including these techniques in future expanded studies.

We believe this clarification strengthens the methodological transparency of the manuscript.

Comment 11: Add a section titled “Limitations and Future Perspectives” at the end of the manuscript.

Response 11: We thank the reviewer for this observation. The ‘Limitations and future perspectives” section (section 4.8.) has been substantially expanded.

Comment 12: Provide a list of abbreviations in alphabetical order at the end of the manuscript.

Response 12: We thank the reviewer for this observation. The abbreviations section has been added at the end of the article, before the references section.

Round 2

Reviewer 2 Report (New Reviewer)

Comments and Suggestions for Authors

The manuscript is well-revised and could be considered for publication

Author Response

We sincerely thank you for the time and effort you invested in reviewing our work. Your initial comments were insightful, constructive and helped us identify areas where additional clarification and detail were needed. We greatly value your perspective and are confident that your guidance has strengthened the overall quality of the manuscript

Point by point response to comments and suggestions for authors

Comment 1: The manuscript is well-revised and could be considered for publication.

Response 1: We thank the reviewer for their time and effort in reviewing our manuscript. We appreciate the insightful comments made and are thankful for their contributions that led to the overall improvement of our manuscript.

Reviewer 3 Report (New Reviewer)

Comments and Suggestions for Authors
  1. Regarding comment 3, the caption of Figure 2 remains unclear. The figure contains five images, yet only four are labeled, and the existing labels appear as “aa, bb, cc, dd,” which is inconsistent and confusing. Please label all panels consistently as (a), (b), (c), (d), and (e), and revise the caption accordingly to ensure clarity for readers.
  2. Regarding Comment 6, please ensure that all table columns report values in the mean ± standard deviation format (X ± Y). There is no need to reference SD in the column headers. For example, in Table S1, the header can simply read “5 min (°C),” and the corresponding data rows can present the values as X ± Y..
  3. Regarding comment 10, Thank you for the response. However, the clarification still does not fully address the core concern behind the request. The recommendation to include SEM, FTIR, and quantitative porosity measurements is not simply an optional enhancement. These analyses are central for understanding the microstructural and chemical changes that can arise from different mixing and cooling techniques in PMMA bone cements. Because the study aims to evaluate how processing affects material performance and thermal behavior, it is essential to demonstrate how the microstructure, porosity distribution, and chemical integrity evolve under these conditions.

Radiographic screening is helpful for detecting macropores and gross defects, but it is not a substitute for microstructural characterization. SEM provides insight into pore morphology, inter-bead fusion, and microcracking. FTIR helps identify potential chemical shifts, polymerization differences, or degradation. Quantitative porosity analysis directly links processing conditions to mechanical and thermal outcomes. Without these, it is difficult for readers to fully interpret the mechanical results or to understand whether the observed effects are driven by structural, chemical, or thermal factors. In its current form, the explanation is brief and does not sufficiently communicate why these characterizations are important for validating the study’s conclusions.

Author Response

Dear reviewer,

We thank you for the time and effort you invested in reviewing our work. Your comments were insightful as always, constructive and helped us identify areas where additional clarification and details were needed. We greatly value your perspective and are confident that your guidance has strengthened the overall quality of our manuscript.

Point by point response to comments and suggestions for authors

Comment 1: Regarding comment 3, the caption of Figure 2 remains unclear. The figure contains five images, yet only four are labeled, and the existing labels appear as “aa, bb, cc, dd,” which is inconsistent and confusing. Please label all panels consistently as (a), (b), (c), (d), and (e), and revise the caption accordingly to ensure clarity for readers.

Response 1: We thank the reviewer for this helpful observation. Figure 2 has been revised to help the reader better understand the labeling. The caption has been modified to match this revision.

Comment 2: Regarding Comment 6, please ensure that all table columns report values in the mean ± standard deviation format (X ± Y). There is no need to reference SD in the column headers. For example, in Table S1, the header can simply read “5 min (°C),” and the corresponding data rows can present the values as X ± Y.

Response 2: We thank the reviewer for this insightful recommendation. The Tables have been revised to include the standard deviation in the X ± Y format. The headers have also been modified, SD columns cut, as they were incorporated into the columns containing the mean values.

Comment 3: Regarding comment 10, Thank you for the response. However, the clarification still does not fully address the core concern behind the request. The recommendation to include SEM, FTIR, and quantitative porosity measurements is not simply an optional enhancement. These analyses are central for understanding the microstructural and chemical changes that can arise from different mixing and cooling techniques in PMMA bone cements. Because the study aims to evaluate how processing affects material performance and thermal behavior, it is essential to demonstrate how the microstructure, porosity distribution, and chemical integrity evolve under these conditions.

Radiographic screening is helpful for detecting macropores and gross defects, but it is not a substitute for microstructural characterization. SEM provides insight into pore morphology, inter-bead fusion, and microcracking. FTIR helps identify potential chemical shifts, polymerization differences, or degradation. Quantitative porosity analysis directly links processing conditions to mechanical and thermal outcomes. Without these, it is difficult for readers to fully interpret the mechanical results or to understand whether the observed effects are driven by structural, chemical, or thermal factors. In its current form, the explanation is brief and does not sufficiently communicate why these characterizations are important for validating the study’s conclusions.

Response 3: We appreciate the reviewer’s emphasis on the importance of SEM, FTIR, and quantitative porosity characterization for understanding the microstructural and chemical effects of different processing conditions in PMMA bone cement. We agree that these techniques provide valuable insight into micropore morphology, polymer network structure and potential chemical shifts.

The present work was designed and approved as a pilot experimental study specifically focused on thermal behavior and compressive strength under two cooling strategies. At this stage of the review process, we are unable to add SEM, FTIR, or quantitative porosity measurements due to time and resource constraints, and because incorporating new analytical methods would require new sample preparation and extended experimental time that exceed the scope and timeline of the current submission.

The central aim of this pilot study is to evaluate how cooling strategies influence polymerization temperature and compressive strength in hand-mixed and vacuum-mixed PMMA. Radiographic screening was used to exclude samples with macroporosity or structural defects, ensuring that the mechanical results reflect the effects of cooling methods rather than gross imperfections. Vacuum mixing followed ISO 5833 standards to minimize porosity variability.

Although microstructural imaging techniques can explain why mechanical changes occur, the current study reports what changes occur in temperature profiles and compressive strength, which remain valid without SEM or FTIR. Our focus is to confirm if the cooling strategies alone lead to measurable and significant differences in the primary clinical outcomes of polymerization and immediate strength, regardless of the underlying microstructural differences.

We believe the current dataset supports the thermo-mechanical conclusions of this pilot study, while recognizing and acknowledging that full mechanistic explanation requires additional microstructural and chemical characterization.

We have revised the manuscript to more acknowledge this limitation and to state that these analyses are essential for mechanistic explanation. We have revised the Limitations section accordingly and clarified the scope of the present work.

Round 3

Reviewer 3 Report (New Reviewer)

Comments and Suggestions for Authors

The explanation for not including SEM, FTIR, and quantitative porosity measurements is not satisfactory. These are fundamental characterization tools for this kind of study, and their absence cannot be justified simply by referring to the “scope of the manuscript.” A research article must present sufficient experimental evidence to support the claims being made, and in the current form, several conclusions are not adequately substantiated. Although your explanation is understandable, it does not meet the standards expected for a rigorous scientific publication. Limiting essential data requirements under the term “scope” is not an appropriate approach for a full research article.

This manuscript is a resubmission of an earlier submission. The following is a list of the peer review reports and author responses from that submission.

Round 1

Reviewer 1 Report

Comments and Suggestions for Authors

Dear Author,

The resubmitted manuscript answered all my questions about the original version. The changes made the article acceptable for publication. I no longer have any questions to clarify.